EMBO
*reports*

# Differential contribution of steady-state RNA and active transcription in chromatin organization

A Rasim Barutcu[1,2,*] (ID), Benjamin J Blencowe[2,3] & John L Rinn[1,†,**] (ID)

## Abstract

Nuclear RNA and the act of transcription have been implicated in nuclear organization. However, their global contribution to shaping fundamental features of higher-order chromatin organization such as topologically associated domains (TADs) and genomic compartments remains unclear. To investigate these questions, we perform genome-wide chromatin conformation capture (Hi-C) analysis in the presence and absence of RNase before and after crosslinking, or a transcriptional inhibitor. TAD boundaries are largely unaffected by RNase treatment, although a subtle disruption of compartmental interactions is observed. In contrast, transcriptional inhibition leads to weaker TAD boundary scores. Collectively, our findings demonstrate differences in the relative contribution of RNA and transcription to the formation of TAD boundaries detected by the widely used Hi-C methodology.

**Keywords** genomic compartment; Hi-C; RNA; topologically associated domains; transcription inhibition

**Subject Categories** Chromatin, Transcription & Genomics; RNA Biology

## Introduction

The precise three- and four-dimensional packaging of chromosomes in the nucleus underlies proper gene expression and in turn cell fate decisions. The genome is folded into structures called topologically associated domains (TADs) and genomic compartments, where transcriptional regulation and DNA replication are regulated [1–3]. Genomic compartments consist of A-type (open) and B-type (closed) compartments. A-type compartments are enriched for active histone marks, a high density of genes, and early replication initiation sites. In contrast, B-type compartments comprise repressive histone marks, and they are associated with a low density of genes and late-replicating

regions [2–4]. Importantly, these structures dynamically change during differentiation and are perturbed in disease [5,6].

Over 30 years ago, RNA was reported to form an integral component of the nuclear matrix [7–10] and has more recently been shown to be involved in the formation of nuclear domains such as the splicing factor-enriched (para)-speckles, the nucleolus, and the Barr body [11–14]. Further suggesting RNA-based roles in nuclear architecture, digestion of RNA, but not of proteins, results in a highly disorganized nucleus as assessed by electron microscopy, and mislocalization of chromatin regulatory complexes [9,15]. Several long non-coding RNAs (lncRNAs), such as FIRRE, XIST, and NEAT1, have important functions in shaping the three-dimensional nuclear structure of DNA [16–19]. Non-coding RNA has been implicated in the formation of loop domains and compartmentalization [20,21]. RNA also has the ability to facilitate the assembly of membrane-less organelles by creating local nuclear environments via liquid–liquid phase separation (LLPS) [22], a mechanism hypothesized to drive higher-order nuclear structures, such as the rRNA-mediated nucleolus formation [23,24]. Finally, RNA can regulate chromatin looping by binding to RNA-binding proteins (RBPs) and chromatin modifiers [17], and can form homo- or heteroduplexes with other RNA molecules essential for RNA processing and chromatin organization [25].

More recently, active transcription has also been postulated as a major driving force in shaping genome architecture [26–29]. Not only was it shown that transcriptional elongation can remodel 3D genome structure [30,31], but transcription can also affect TAD interactions [26,32]. However, whether it is structural roles involving pre-existing nuclear RNA content—or the processes tied to active transcription—that more strongly impact higher-order nuclear architecture on a genome-wide scale, as detected by Hi-C, has not been previously determined.

Here, we investigate the role of total cellular, steady-state single-stranded RNA on genome architecture by RNase treatment before and after crosslinking cells followed by Hi-C analysis. Furthermore, to assess the effects of the act of transcription on genome organization, we performed Hi-C in the presence of transcriptional inhibitor. Surprisingly, we find that TAD boundaries remain well preserved upon RNase treatment. However, their strength, as measured by the

1  Department of Stem Cell and Regenerative Biology, Harvard University, Cambridge, MA, USA
2  Donnelly Centre, University of Toronto, Toronto, ON, Canada
3  Department of Molecular Genetics, University of Toronto, Toronto, ON, Canada
   *Corresponding author. Tel: +1 416 978 7150; E-mail: rasim.barutcu@utoronto.ca
   **Corresponding author. Tel: +1 303 735 5059; E-mail: john.rinn@colorado.edu
   †Present address: Department of Biochemistry, BioFrontiers, University of Colorado, Boulder, CO, USA

degree of interactions that occur across a TAD boundary, is significantly decreased upon transcriptional inhibition, compared to the effects observed with RNase treatment. Taken together, these results suggest different relative contributions of the role of active transcription and steady-state nuclear RNA in the formation of TAD boundaries and genomic compartments.

# Results

### Experimental strategy and characterization of RNase treatment and transcriptional inhibition

We investigated the global role of RNA in maintaining genome topology by performing Hi-C in the presence and absence of RNase A treatment of human K562 cells. This treatment was optimized to digest all single-stranded regions of steady-state RNA, and resulted in RNA fragments of < 500 nucleotides (Fig 1A and B). RNase A digestion, or treatment with RNase inhibitors (as a control), was performed either before or after formaldehyde crosslinking. These treatments are referred to below as "bXL" for RNase treatment *before crosslinking*, and "aXL" for RNase treatment *after crosslinking*, respectively. The *in situ* Hi-C protocol was then performed on these samples [33] (Fig 1A). RNA agarose gel analysis confirmed total RNA depletion following RNase A treatment, whereas RNase H or RNase inhibitor-treated samples, and mock controls, show prominent levels of 18S and 28S ribosomal RNA, indicative of intact RNA (Fig 1B).

Previous studies have provided evidence that the act of transcription may play a role in shaping chromatin organization [26,27,29,30,34,35], yet the global consequences of complete transcriptional arrest (i.e., blocking all three polymerases) on mammalian chromosomal architecture are not well understood. Accordingly, to assess whether genome organization is affected differently by the act of transcription compared to loss of steady-state RNA following RNase treatment, we treated K562 cells with actinomycin D (ActD), a drug that quickly (i.e., within 24 h) and irreversibly inhibits all polymerases and results in complete transcriptional arrest [36–38] (Fig 1C). Agarose gel electrophoresis analysis of total RNA extracted from the same number of cells as controls upon 24 h of ActD treatment revealed that ~50% of the 28S/18S ribosomal RNA bands remain, indicating the inhibition of global transcription and a decrease in nuclear RNA content (Fig 1D). Furthermore, we performed qRT–PCR on representative genes (PTEN, FNDC3B, and STAM) that previously were shown to be sensitive to transcriptional inhibition in K562 cells [39]. Consistently, these genes display significant (*t*-test, $P < 0.05$) decreases upon 24 h of ActD treatment (Fig 1E), further confirming inhibition of transcription.

RNase treatment or transcriptional inhibition has been reported to result in morphological changes in the nucleus [40,41]. We therefore sought to assess the consequences of RNase and ActD treatments on overall cell integrity and nuclear morphology in several ways. First, we addressed whether Tween-20 permeabilization or RNase A treatment leads to apoptosis of the cells by performing staining with cleaved caspase 3, an apoptotic marker. The treated cells show weak but comparable levels of caspase 3 signal compared with the control treatments indicating that Tween-20 permeabilization and RNase A treatment do not lead to extensive apoptosis within the time frame of the experiment (Fig EV1A). Second, to

investigate the effects of RNase and ActD treatments on nuclear domain architecture, we visualized nucleolar morphology by performing immunostaining with an antibody to Fibrillarin, which concentrates in nucleoli (Fig EV1B). While K562 cells display a range of nucleolar sizes, we observe similar patterns of Fibrillarin immunostaining between the RNase A-treated and control cells (Fig EV1B). Finally, DAPI staining of K562 cells, as well as of HeLa cells, under the same treatment conditions as described above, reveals that cells prior to crosslinking appear to have normal nuclear morphology (Fig EV1B), regardless of whether or not they were treated with RNase. Tween-20 permeabilization alone prior to crosslinking does not result in significant differences in nuclear size compared with cells treated with RNase A after permeabilization in K562 cells (Fig EV1B–D).

Taken together, these data suggest that overall nuclear structure is largely preserved when cells are cross-linked prior to RNase and ActD treatments.

### RNA depletion results in reduction in B-type compartmental interactions

In order to globally assess the changes in chromatin architecture, we performed *in situ* Hi-C analysis of K562 cells following RNase treatment and ActD inhibition, with and without prior crosslinking, as described above, and with two biological replicates (Materials and Methods, Table EV1). The Hi-C profiles of the replicate 1st eigenvalues, which is a measure of genomic compartmentalization, display higher correlations with each other (average Pearson correlation $R^2 = 0.9$, see Materials and Methods, Fig EV2A and B) than with the other samples. This observation demonstrates high consistency between the preparations and differential effects of the RNase and ActD treatments.

We next analyzed the Hi-C interaction matrices for A-type and B-type compartmentalization. As described previously [2], the Hi-C heatmaps reveal two unique patterns of interactions, which have been identified either as euchromatic, gene-rich, and highly transcribed (A-type compartments) or as heterochromatic, gene-poor, and silent (B-type compartments). A Pearson correlation analysis of Hi-C heatmaps reveals more *intra*-compartmental than *inter*-compartmental associations, or in other words, the A and B compartments associate more frequently within themselves, rather than with each other. When a principal component analysis (PCA) is used to separate these Hi-C heatmaps, the first eigenvector is used to capture this "plaid-pattern", where the positive values represent the A-type compartment and the negative values represent the B-type compartment. Comparison of the bXL or aXL samples revealed that the bXL RNase A sample had a dramatically reduced "plaid-pattern" compared with the aXL RNase A sample, when compared to controls (Fig 2A and B).

Next, we plotted the Pearson correlation matrices, where the red color denotes A-type and the blue color denotes B-type compartmentalization. This analysis reveals that while there is no appreciable change between the aXL CTRL and aXL RNase A-treated samples, the bXL RNase A sample shows a significant reduction in Pearson correlations when compared to the bXL CTRL sample (Figs 2C and D, and EV3A). We then analyzed the switching of open (A-type) to closed (B-type) compartments, and vice versa, and observed that the majority (> 90%) of the compartments are stable in samples treated with RNase either before or after crosslinking (Fig EV3B and C).

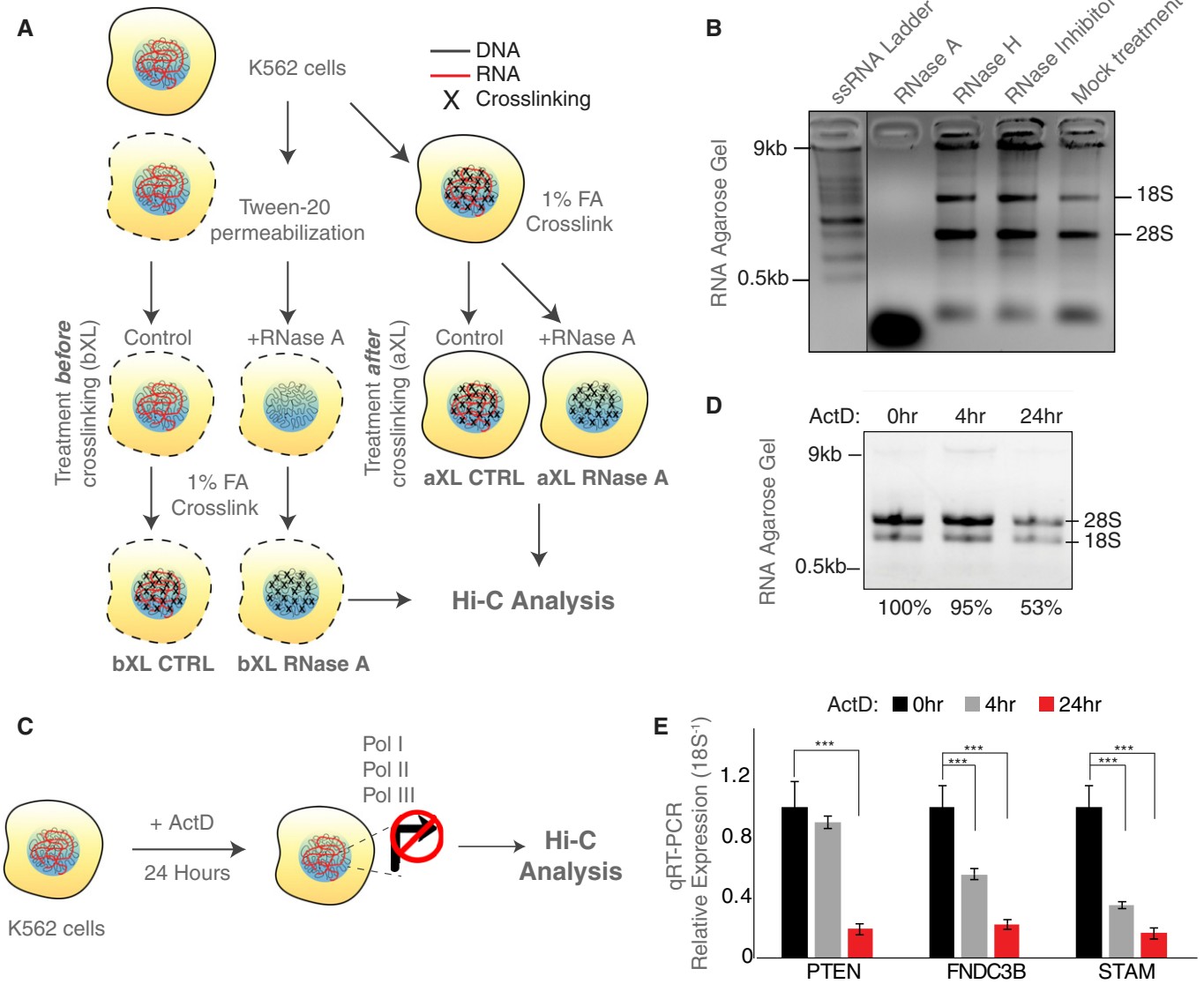

**Figure 1. Overall genome organization is preserved upon RNase digestion.**

A   Experimental scheme of RNase A digestion of K562 cells before and after formaldehyde (FA) crosslinking (termed as bXL and aXL, respectively) followed by the Hi-C assay.

B   Agarose gel electrophoresis of RNA extracted from RNase-treated or control K562 cells.

C   Experimental scheme of ActD treatment of K562 cells followed by Hi-C analysis.

D   Agarose gel electrophoresis of RNA extracted from control, and 4- and 24-h ActD-treated K562 cells.

E   Barplots showing qRT–PCR gene expression levels (mean ± SD) of PTEN, FNDC3B, and STAM genes in control, and 4- and 24-h actinomycin D-treated K562 cells. P-value: two-tailed Student's t-test, (***$P < 0.05$). The results represent data from three technical replicates from three independent biological preparations.

Plotting the $1^{st}$ eigenvalues revealed a perturbation of the B-type compartments upon treatment with RNase before crosslinking (Fig 2E), even though the compartment switching (i.e., A to B, or B to A) was minimal upon RNase treatment (Fig EV3B and C).

To quantify the compartmental interactions, we generated saddle plots by ranking the $1^{st}$ eigenvalues, then binning them into 30 quantiles, followed by calculating the pairwise distance-normalized observed/expected interaction frequencies among the 30 bins. We generated these plots both for the *cis*-contacts and for the *trans*-contacts (Fig 2F). We determined a reduction in compartmental

interactions, especially B-B interactions, in the saddle plots generated with both in-*cis* and in-*trans* chromosomal data (Fig 2F).

Taken together, these results suggest that RNase treatment before crosslinking leads to disruption of especially the B-type compartmental interactions.

## RNA depletion does not affect TAD boundaries

It has been proposed that TAD organization is primarily driven by a loop extrusion complex [42–44]. In light of this current

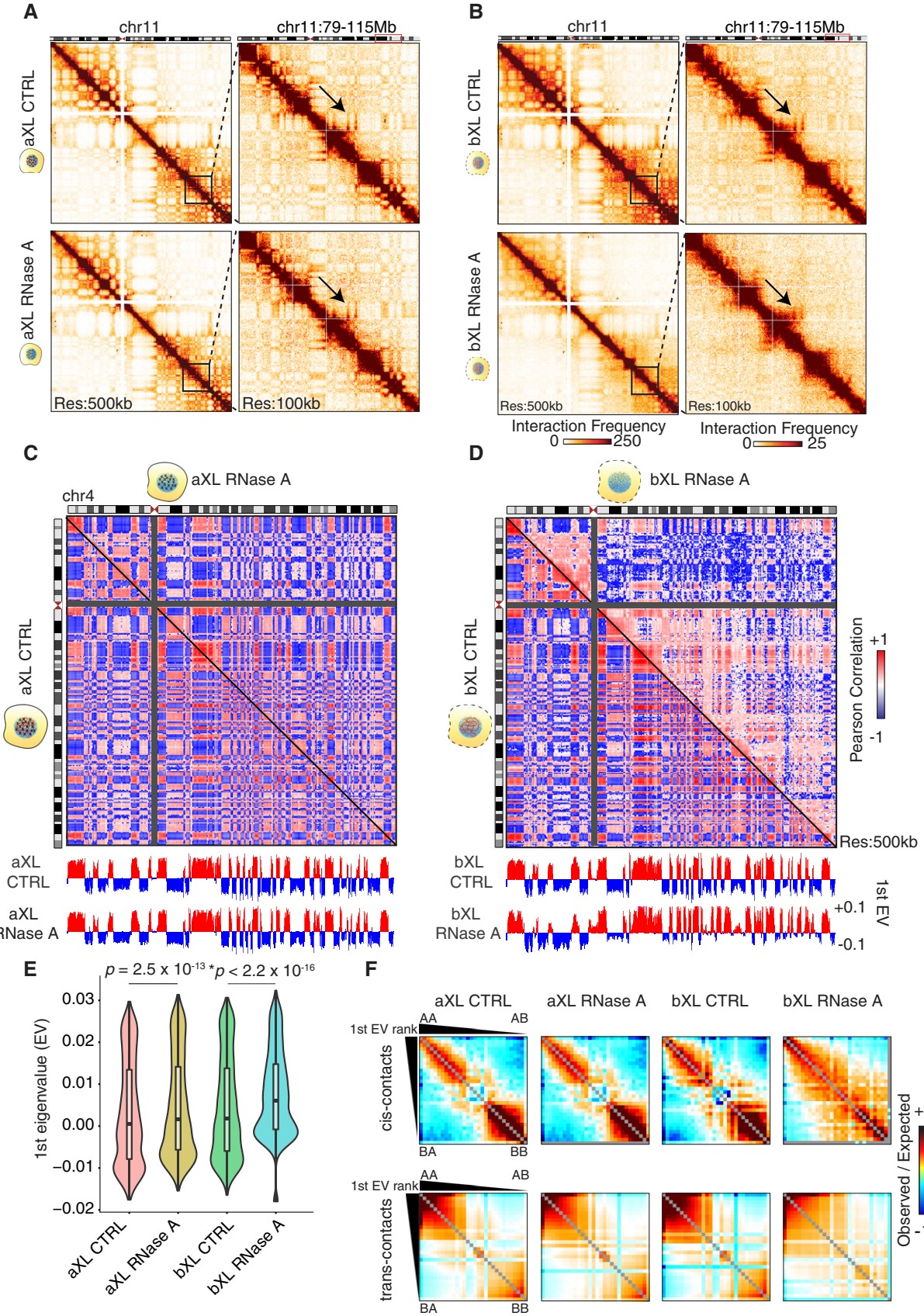

**Figure 2. RNA depletion results in reduction in B-type compartmental interactions.**

A  Hi-C interaction heatmaps at 500-kb (left) resolution showing all of chromosome 11, and at 100-kb resolution (right) showing chr11: 79–115 Mb for aXL control and RNase A-treated cells. The arrows represent similar patterning of long-range interactions in samples treated with RNase after crosslinking.

B  Hi-C interaction heatmaps at 500-kb (left) resolution showing all of chromosome 11, and at 100-kb resolution (right) showing chr11: 79–115 Mb for bXL control and RNase A-treated cells. Black arrow indicates the reduction in long-range interactions in the bXL RNase A dataset. The arrows represent perturbed patterning of long-range interactions in samples treated with RNase before crosslinking.

C, D  Pearson correlation heatmaps showing the genomic interactions at 500-kb resolution for chromosome 4, with the 1$^{st}$ principal components (1$^{st}$ PC) below for aXL (C) and bXL (D) samples. The bXL RNase A sample shows reduced Pearson correlations across the genome.

E  Violin plot showing the first eigenvalues (calculated by the *trans*-chromosomal data) in control and RNase-treated cells before and after crosslinking. The bXL RNase A sample shows reduction in the negative eigenvalues, indicative of perturbation of B-type compartments. *P*-value: Wilcoxon rank-sum test. In the violin plots, the horizontal bands represent the median, the error bars represent ± 1.5 × IQR values, and the outer shapes represent the density of the data points. The figures represent data generated from the pooled Hi-C datasets with two biological replicates.

F  Saddle plots showing the compartmental interactions for both *cis*- and *trans*-contacts across the conditions. The bXL RNase A samples display reduced B-B-type interactions.

mechanistic model, we next asked whether TAD structures are preserved in the absence of single-stranded RNA and analyzed the TAD boundaries by using the insulation method, which calculates the average Hi-C interaction frequency of a sliding window (insulation plot) and detects TAD boundaries based on the decreased scores of the insulation plot [45]. We find that TAD structures are similarly present in each dataset, and the log2 ratios of interaction frequencies between RNase-treated versus control samples do not reveal any specific patterns (Fig 3A). Moreover, although TAD boundary scores are reduced upon initial Tween-20 permeabilization, the aXL and bXL controls compared to the RNase A treatments do not show a significant difference in these scores (Fig 3B). Consistent with these findings, plotting the mean interaction frequency within ± 1 Mb of all of the TAD boundaries, as well as the RNase A/CTRL log2 ratios, also reveals similar interaction profiles between the datasets (Fig 3C and D). Altogether, these results suggest that TAD formation is not appreciably affected by the loss of RNA in either the aXL and bXL samples upon RNase A treatment.

**Genomic compartments are preserved upon transcriptional inhibition**

We then analyzed the TADs and genomic compartments upon transcriptional inhibition. Interestingly, calculating the Pearson correlation of the Hi-C interaction matrices reveals that, similar to the bXL RNase A samples (Fig 2C and D), 24-h ActD-treated cells show reduced Pearson correlations of genomic interactions (i.e., plaid-patterning) compared with controls (Fig 4A and B). However, a principal component analysis of the Pearson correlation heatmaps reveals preservation of the A-type and B-type compartmentalization, as the majority (> 90%) of the compartments appear to be similar (Figs 4C and EV3D). Quantification of compartmental interactions by saddle plots showed overall similar profiles between the control and 24-h ActD-treated cells. However, there was a slightly higher rate of inter-compartmental interactions upon transcriptional arrest (Fig 4D). Next, to further investigate whether the underlying chromatin structure is similar at the compartmental level, we plotted the interaction frequency as a function of genomic distance, which revealed similar decay rates at distances > 2 Mb between control and 24-h ActD-treated cells (Fig 4E). These results suggest the preservation of the genomic compartments upon transcriptional inhibition, more so than bXL RNase treatments.

**Transcriptional inhibition results in weakening of TAD boundary strength**

It was previously shown in *Drosophila* that RNA Pol II inhibition leads to changes in TAD dynamics [26]. Next, we analyzed whether TAD structures are altered upon transcriptional inhibition. Visualization of TADs at 40-kb resolution indicates that the overall structure of TADs is not perturbed. However, there is an increased rate of interactions across the TAD boundaries, confirming previous reports (Fig 5A) [26]. Consistent with this, the TAD boundary scores (calculated by the insulation score method [45]) of 24-h ActD-treated cells are significantly reduced (Wilcoxon rank-sum test, $P < 2.2 \times 10^{-16}$), as compared to controls (Fig 5B). We next plotted the average Hi-C interaction frequency ± 1 Mb of all TAD boundaries detected in control and 24-h ActD-treated datasets. Compared to RNase/control datasets (Fig 3C and D), a pronounced weakening of TAD boundaries upon transcriptional inhibition is observed, specifically, when the log2 ratio of 24-h/0-h control interaction profiles around the TAD boundaries is plotted (Fig 5C). Moreover, this analysis reveals an increased rate of interactions across, but not within the TAD boundaries. We then performed an aggregate peak analysis [46], which measures the aggregate enrichment of putative peaks in a Hi-C contact matrix, based on the K562 loop coordinates defined in a previous high-resolution Hi-C study [4]. Consistent with the weakening of TAD boundaries, there was a ~50% decrease in loop intensity as assessed both by z-score calculation and by the log2 fold change (Fig 5D). Therefore, this result suggests that the act of transcription has a more dramatic effect on TAD boundaries than RNase treatment.

Next, we sought to ask whether CTCF binding affected the reduction in TAD boundary scores upon 24 h of ActD treatment. It was previously shown that the higher number of insulators bound at TAD boundaries is associated with higher insulation scores (i.e., stronger boundaries) [26]. To address this question, we intersected the ENCODE K562 CTCF ChIP-seq data with ActD Hi-C data and categorized the 40 kb TAD boundary bins by the number of CTCF sites bound (1 or less, 2 or more) (Fig 5E). As expected from previous studies, we observed significantly higher scores of TAD boundaries that harbored 2 or more CTCF sites in both control and 24-h ActD-treated cells (Fig 5E). Interestingly, ActD treatment led to a significant decrease in TAD boundary scores, when compared to controls, regardless of the number of CTCF sites bound (Fig 5E), suggesting that the act of transcription may have roles at TAD boundaries independent of CTCF. Consistent with these results, plotting the interaction frequency as a function of genomic distance of interactions at distances less than

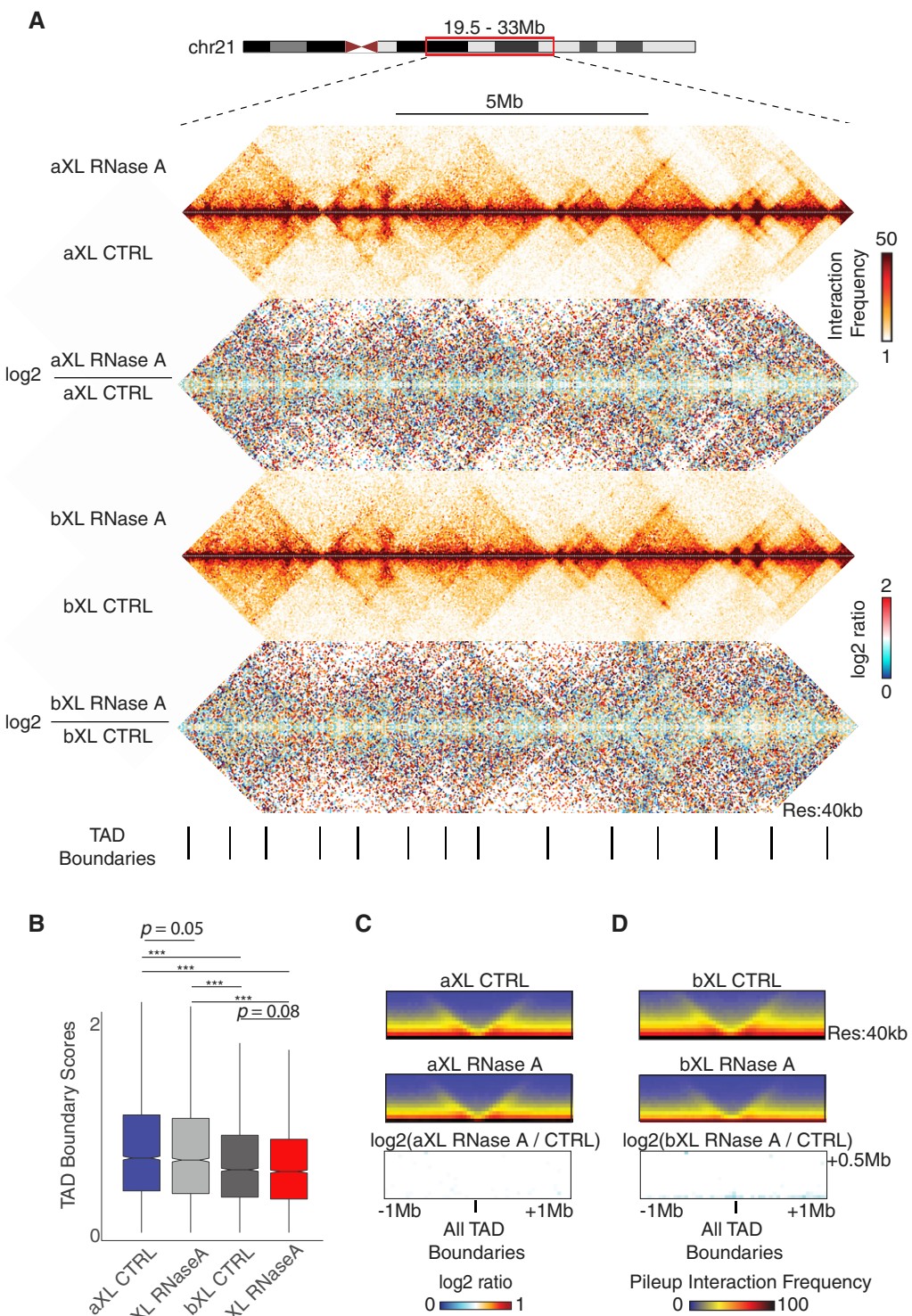

**Figure 3. RNA depletion does not affect TAD boundaries.**

A    Interaction heatmaps at 40-kb resolution zooming in chr21: 19.5–33 Mb and showing the TAD structures in aXL and bXL datasets. The aXL and bXL RNase A/control log2 ratios are shown below the heatmaps.

B    Boxplot showing the TAD boundary scores for the aXL and bXL datasets. *P*-value: Wilcoxon rank-sum test (\*\*\**P* < 0.05). The horizontal bands in the boxplots represent the median, the error bars represent ± 1.5 × IQR values, and the outer shapes represent the density of the data points. The figures represent data generated from the pooled Hi-C datasets with 2 biological replicates.

C, D    Mean interactions frequencies ± 1 Mb centered on all TAD boundaries for each dataset at 40-kb resolution, as well as the log2 ratios of aXL (left panel) or bXL (right panel) samples. The TAD structures are not altered upon RNase A treatment in samples treated with RNase after crosslinking.

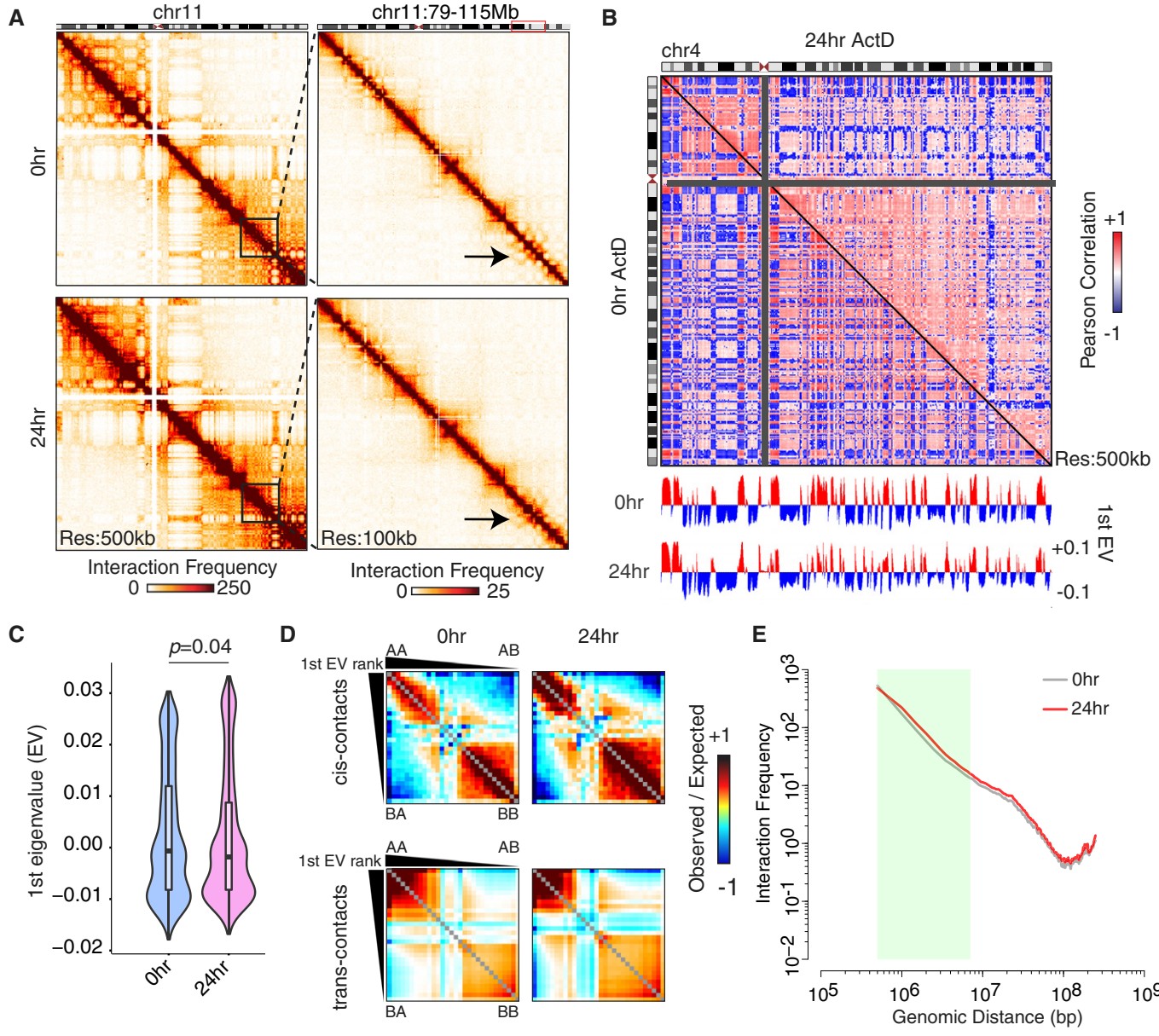

**Figure 4. Genomic compartments are largely preserved upon transcriptional inhibition.**

A   Hi-C interaction heatmaps at 500-kb (left) resolution showing all of chromosome 11, and at 100-kb resolution (right) showing chr11: 79–115 Mb for control and 24-h ActD-treated cells. The arrows point out to perturbed long-range interactions at shorter scales.

B   Pearson correlation heatmaps showing the genomic interactions at 500-kb resolution for chromosome 4, with the 1st principal component (1st PC) below for control and 24-h ActD-treated cells.

C   Violin plot showing the first eigenvalues (calculated by the *trans*-chromosomal data) in control and ActD-treated cells. *P*-value: Wilcoxon rank-sum test. In the violin plots, the horizontal bands represent the median, the error bars represent $\pm$ 1.5 $\times$ IQR values, and the outer shapes represent the density of the data points. The figures represent data generated from the pooled Hi-C datasets with 2 biological replicates.

D   Saddle plots showing the compartmental interactions for both *cis*- and *trans*-contacts across the conditions.

E   Scaling plot generated by 500-kb resolution Hi-C data showing the interaction frequency as a function of genomic distance.

3 Mb reveals a differing decay rate of interactions in 24-h ActD-treated cells when compared to controls (Fig 5F).

Collectively, our data suggest that the act of transcription has a more predominant role than depletion of steady-state RNA in the organization of TAD borders, and vice versa for A- and B-type compartment organization.

## Discussion

Here, we have investigated the relative contribution of the act of transcription and total nuclear steady-state RNA content on the organization of nuclear chromatin by performing Hi-C on RNase-treated cells before and after crosslinking, as well as upon

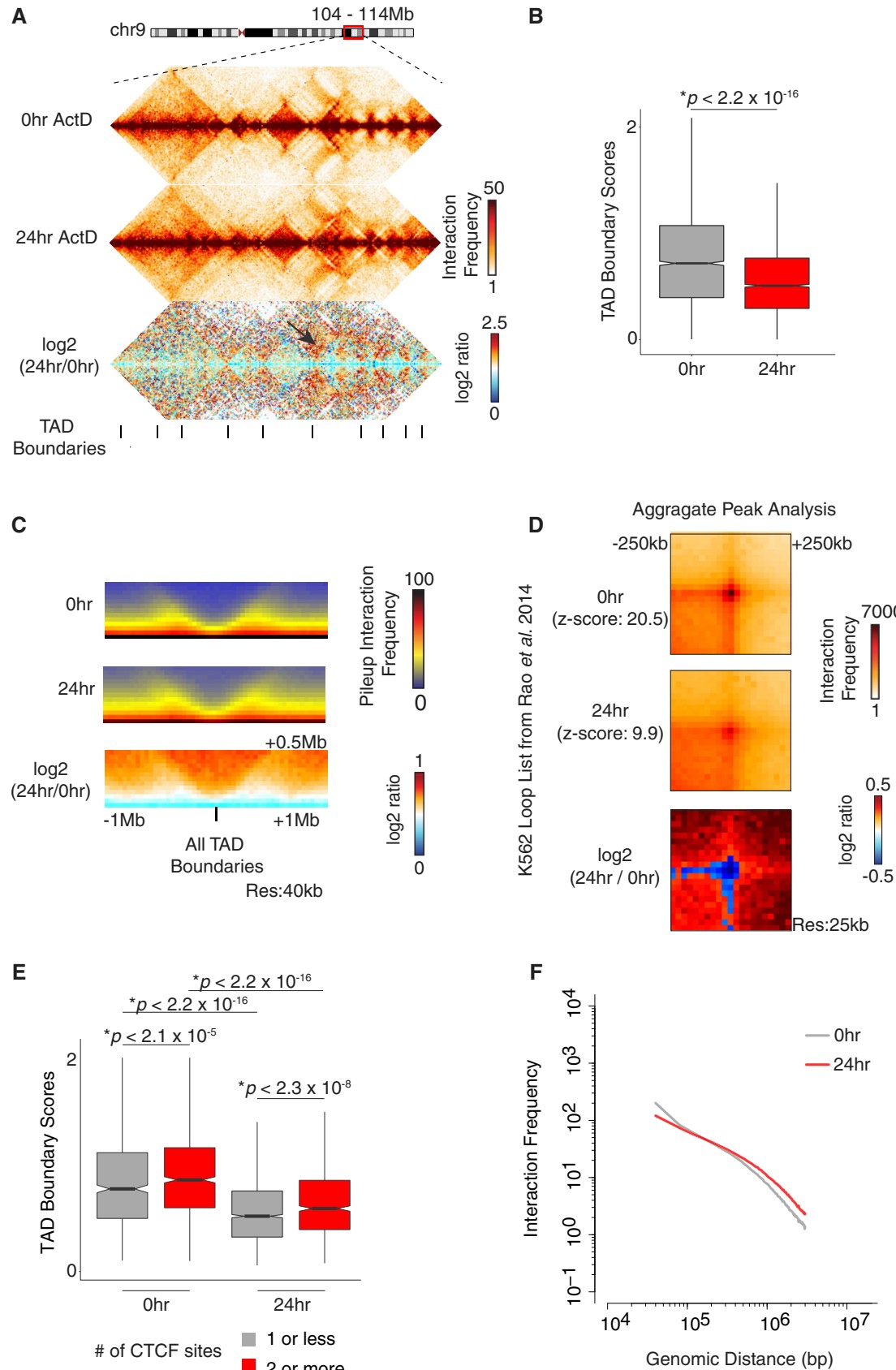

**Figure 5.**

**Figure 5.  Transcriptional inhibition results in weakening of TAD boundary strength.**

A   Interaction heatmaps at 40-kb resolution zooming in chr9: 104–114 Mb and showing the TAD structures, as well as the log2 ratios, in control and transcriptionally inhibited datasets. Black arrow indicates the increase in *inter*-TAD interactions.

B   Boxplot showing the TAD boundary scores in control and 24-h ActD-treated cells. *P*-value: Wilcoxon rank-sum test. The horizontal bands in the boxplots represent the median, the error bars represent ± 1.5 × IQR values, and the outer shapes represent the density of the data points. The figures represent data generated from the pooled Hi-C datasets with 2 biological replicates.

C   Mean interaction frequencies centered on all TAD boundaries ± 1 Mb for each dataset at 40-kb resolution, as well as the log2 ratios. The *inter*-TAD interactions are increased upon transcriptional inhibition.

D   Aggregate peak analysis based on the K562 loops coordinates identified in [4]. The heatmaps are centered on the loop positions ± 250 kb using 25-kb resolution Hi-C data. There is a ~50% reduction in loop intensity as assessed by z-scores and log2 fold change.

E   Boxplot showing the TAD boundary scores in control and 24-h ActD-treated cells for boundaries that are bound with differing numbers of CTCF binding sites. The reduction in TAD boundary scores upon ActD treatment is independent of CTCF binding. *P*-value: Wilcoxon rank-sum test. The horizontal bands in the boxplots represent the median, the error bars represent ± 1.5 × IQR values, and the outer shapes represent the density of the data points. The figures represent data generated from the pooled Hi-C datasets with 2 biological replicates.

F   Scaling plot generated by 40-kb resolution Hi-C data showing the interaction frequency as a function of genomic distance, with upper distance limit of 3 Mb. The 24-h ActD-treated samples show a slower rate of decay at shorter distances, correlating with the reduction in insulation scores.

transcriptional inhibition. Collectively, the findings in this study have several important implications (Fig 6).

Our findings suggest that the pool of RNA in the cell appears to be largely dispensable for the maintenance of TADs, despite evidence of lncRNAs and other RNA species facilitating TAD architecture [20,47,48]. Treating cells with RNase before and after crosslinking resulted in near-identical patterns of TADs. This implies that while pre-existing transcribed RNA may play a role at small local scales (i.e., within a TAD), or mediate *inter*-chromosomal interactions [17,49], overall it does not appear to significantly influence the TAD boundary formation feature of nuclear architecture. A mechanistic model, known as the loop extrusion model, has been proposed for TAD formation [4,43,48,50]. Our finding that TAD boundaries remain intact in cells treated with RNase A, either before or after formaldehyde crosslinking (Fig 3), is consistent with a key feature of this model, namely that TAD formation is primarily driven by DNA–protein and protein–protein interactions rather than by RNA [51].

On the other hand, transcriptional inhibition, when compared to the effects of RNase treatment, results in a more pronounced weakening of TAD boundaries. We do note that the weakening of TAD boundaries by transcriptional inhibition may be due to decreased transcript and protein levels of the loop extrusion complex (e.g., CTCF and cohesin). Interestingly, the weakening of TAD boundaries was independent of CTCF binding (Fig 5E), suggesting supporting roles for transcription and CTCF in the formation of TADs and regulation of *inter-TAD* interactions. This finding is consistent with previous reports that transcriptional inhibition as well as transcriptional elongation can displace cohesin from CTCF sites and disrupt chromatin interactions [26,30,32,52], correlating with weakening of TAD boundaries.

Furthermore, our results are consistent with the hypothesis that TAD boundaries form by the loop extrusion complex [53,54], and

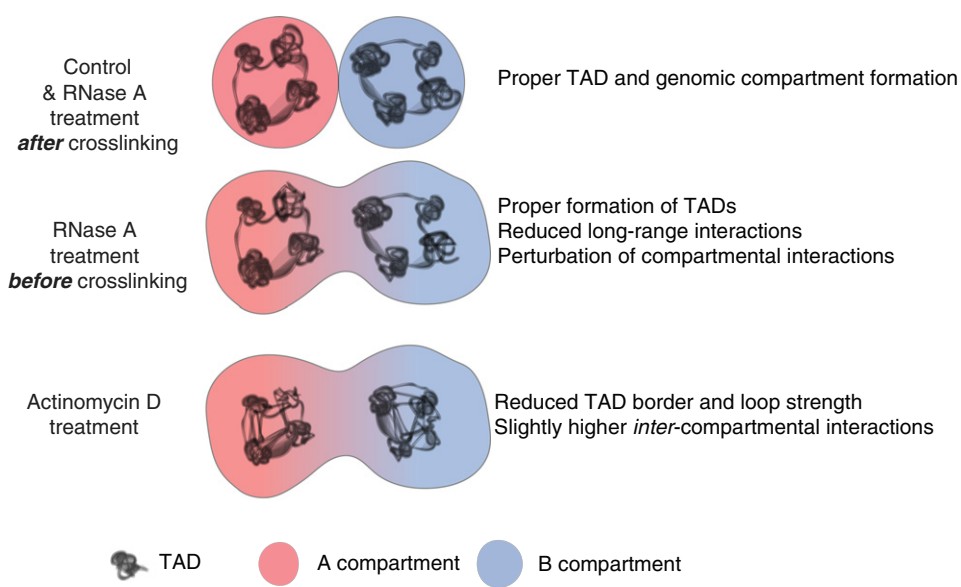

**Figure 6.  Model summarizing the nuclear structure changes observed upon different treatments.**

RNase treatment of cells before crosslinking results in reduction in long-range interactions specifically at B-type compartments, but does not significantly alter TAD boundaries. On the other hand, transcriptional inhibition leads to significant weakening of TAD boundaries without drastic alterations in genomic compartmentalization.

their fine-tuning (i.e., regulation of specific enhancer–promoter interactions or *inter*-TAD interactions) may be regulated by an interplay between loop extrusion and the act of transcription, in addition to the selective binding of transcription factors and chromatin modifiers at TAD boundaries [1,55–57]. However, it is worth noting that Vian and colleagues have demonstrated that TADs can disappear upon cohesin degradation and reappear after rescue, even in the absence of transcription [32]. Similarly, previous reports suggested that the act of transcription can influence the 3D structure of the nucleus [27,58–60]. Indeed, although the compartmentalization was not markedly altered (Fig 4) we observed a reduction in TAD boundary strength upon transcriptional inhibition (Fig 5). These results are concordant with a model where the act of transcription and or its product influences the nuclear positioning of the newly transcribed DNA locus via interaction with nuclear matrix proteins [27,28,61,62].

Lastly, the RNase A-treated cells before crosslinking display a subtle perturbation of compartmental interactions, especially in B-type compartments, even though TAD structures were preserved (Figs 2 and 3). We speculate that the reduction in B-type compartmentalization may represent a yet unidentified function for pre-existing and newly transcribed RNA in genomic compartmentalization, possibly through LLPS, as it was hypothesized that the heterochromatin may be formed by an LLPS mechanism [23]. Thus, our findings support a role for RNA and transcription in regulating different levels of genome organization. Previously, RNase treatment has been associated with the disruption of some, but not all, nuclear bodies (Fig EV1) [63], as well as nuclear changes in both euchromatin and heterochromatin organization [40,41]. What further remains to be resolved is whether RNA influences specific *inter*-chromosomal interactions observed at mRNA [64] and lncRNA loci [17,49,65], a feature not readily observed by Hi-C approaches at high resolutions [66].

In summary, our findings provide insight into how total cellular RNA and the act of transcription contribute differently to the establishment of genome architecture (Fig 6). Our results suggest that once formed, the act of transcription has a more critical role in the integrity of TAD boundaries than pre-existing steady-state RNA. A surprising result is that overall genome organization, as assessed by Hi-C, is not substantially altered by the loss of RNA after either treatment, even though perturbed nuclear organization is observed by microscopy (Fig EV1) [67]. Further molecular and detailed microscopy and molecular experiments, which specifically investigate the mechanism and effects of RNase treatment on nuclear structure and binding of chromatin modifiers, will be critical to unravel the underlying principles of higher-order genome architecture.

## Materials and Methods

### Cell culture

K562 cells were obtained from ATCC (#CCL-243) and cultured in RPMI 1640 (Thermo Fisher #22400-105), 10% fetal bovine serum (Thermo Fisher #26140-079), 1% L-glutamine (Thermo Fisher #25030-164), and 1% penicillin/streptomycin (Thermo Fisher #15140-163) in a 37°C incubator. The media were changed every 2 days, and when the cells' density reached ~8 × 10$^5$ cells/ml, the culture was split to ~3 × 10$^5$ cells/ml with fresh, warm media. One of three sets of K562 cultures was mock-treated (0 h), and the other two sets were treated with actinomycin D (5 μg/ml) (24 h), and all 3 sets of cultures were incubated for 24 h and harvested for Hi-C crosslinking. The set which was mock-treated was regarded as 0 h, and the two other sets as 24 h of treatments. HeLa cells were obtained from ATCC (#CCL-2) and cultured in Dulbecco's modified Eagle's medium (DMEM) (Sigma-Aldrich #D5796) with 10% fetal bovine serum and 1% penicillin/streptomycin.

### Permeabilization and crosslinking of cells

For the RNase A treatment before crosslinking (bXL) condition, ~10 × 10$^6$ K562 cells were aliquoted and spun down at 1,000 × *g* for 10 min at 4°C. The supernatant was removed, and the cell pellet was re-suspended with 1 ml of cold PBS + 0.02% Tween-20 followed by incubation on ice for 10 min. Next, either of 35U of RNase A (Qiagen #19101) or of 30U of RNase inhibitor (NEB, #M0314L) was added to the suspension and incubated at 37°C for 30 min in a thermomixer at 950 rpm. Then, the cells were spun down at 1,000 × *g* for 10 min at 4°C, washed once with 1 ml cold PBS with protease inhibitor cocktail (Roche, #04693159001), and then re-suspended in 10 ml 1× PBS at room temperature with the protease inhibitor cocktail, followed by crosslinking of the cells with formaldehyde at 1% final concentration and quenching the formaldehyde with glycine. Then, the RNase A and RNase inhibitor-treated samples were processed for downstream Hi-C experiments. For the RNA agarose gel figures in Fig 1, RNA extraction was performed using the TRIzol reagent (Thermo Fisher #15596026), which inhibits RNase during sample homogenization and extraction [68,69].

### Microscopy

K562 cells were cultured on coverslips that were pre-coated with Poly-L-lysine solution (Sigma-Aldrich #P8920) on a 6-well plate. The adherent HeLa cells were cultured on non-coated coverslips. The cells were then washed with 1× PBS, Tween-20 permeabilized and RNase A or RNase inhibitor treated as described above, fixed with 4% paraformaldehyde, and stained with either 1:400 dilution of anti-cleaved caspase-3 antibody (Cell Signaling #9664), or 1:500 dilution of anti-Fibrillarin antibody (Abcam #4566), followed with the DAPI staining (Thermo Fisher #D1306). The HeLa cells were imaged on a Zeiss spinning disk confocal microscope. Since K562 cells are smaller in size, we imaged these cells using the LSM880 with the Airyscan (Zeiss) system, which enhances the sensitivity and resolution beyond the diffraction limit of light. We imaged the whole cell by *z*-stacks (with 0.45-μm slices) and present the maximum intensity projection of the images. The nuclear area of the nuclei in Fig EV1 was quantified by using the ImageJ (Fiji) software [70].

### qRT–PCR analysis

RNA was extracted by using TRIzol Reagent (Thermo Fisher #15596026) and cDNA was generated by using the SuperScript III Reverse Transcriptase Kit (Thermo Fisher 18080044), according to manufacturer's instructions. The qRT–PCR data were analyzed by

using the $2^{(-\text{delta/delta Ct})}$ method. The primer sequences used for the qRT–PCR experiments are listed below.

| | |
|---|---|
| hPTEN_qPCRFW | TGGATTCGACTTAGACTTGACCT |
| hPTEN_qPCRRV | GGTGGGTTATGGTCTTCAAAAGG |
| hFNDC3B_qPCRFW | TCTCGTTCAAGTTAATCCAGGTG |
| hFNDC3B_qPCRRV | ACATGGCTGAGGGGTAGCTT |
| hSTAM_qPCRFW | AATCCCTTCGATCAGGATGTTGA |
| hSTAM_qPCRRV | CGAGACTGACCAACTTTATCACA |
| hTLE4_qPCRFW | CGACCTGAGCAAGATGTACCC |
| hTLE4_qPCRRV | CGATCACAGGATTCGGAAATTGT |
| h18S_qPCRFW | GGCCCTGTAATTGGAATGAGTC |
| h18S_qPCRRV | CCAAGATCCAACTACGAGCTT |

### In situ Hi-C

Hi-C libraries were generated with the *in situ* ligation using the *HindIII* restriction enzyme [33], Briefly, after crosslinking the cells with 1% formaldehyde for 10 min at room temperature, the nuclei were extracted by dounce homogenization. Next, the chromatin was digested with *HindIII* either in the presence of 35U of RNase A (Qiagen #19101) or in the presence of 30U of RNase inhibitor (NEB, #M0314L). These samples that are treated after formaldehyde crosslinking (aXL) are referred as "aXL RNase A" and "aXL CTRL", respectively. The bXL samples, after treatment with RNase or RNase inhibitors as described above, were processed according to the *in situ* Hi-C protocol [33]. The digested chromatin was then end-labeled with biotin-14-dCTP (Thermo Fisher 19519016), and *in situ* ligation was performed. Extraction of the DNA was performed by phenol–chloroform extraction, and the biotin was removed from unligated ends, followed by shearing of the DNA by using a Covaris S220 instrument to 100–500 bp size range. After A-tailing, biotin pull-down, and adapter ligation, paired-end sequencing was performed on a HiSeq instrument. Hi-C libraries for each condition were generated in two replicates, which showed high 1st eigenvector correlation (average Pearson's correlation $R^2 = 0.9$).

### Analysis of Hi-C datasets

Hi-C mapping, filtering, correction, and binning were performed with the HiC-Pro software v2.8.1 [71]. The sequenced reads were mapped to the hg19 human reference genome, and corrected for biases, including read depth [71]. There was a high correlation among all the Hi-C biological replicates, as assessed by the first eigenvalues, indicating the high quality and reproducibility of the datasets. Therefore, we pooled all biological replicates for each condition and processed them as a single Hi-C dataset, which yielded a depth sufficient enough to assess large-scale genomic structures such as genomic compartments and TADs [72,73].

### Insulation and TAD boundary analysis

TAD analysis was performed by using the "Insulation Method" by a publicly available script (matrix2insulation.pl) [45]. The following

options were used to call the TAD boundaries using the 40-kb resolution Hi-C interaction heatmaps: "–is 480000 –ids 320000 –im iqrMean –nt 0 –ss 160000 –yb 1.5 –bmoe 0 –bg". The script is available through GitHub (https://github.com/dekkerlab/cworld-dekker).

### Genomic compartment analysis

Genomic compartments were calculated by using genome-wide *intra*-chromosomal Hi-C heatmaps at 500-kb resolution. The compartments were calculated by using the "matrix2compartments.pl" script available via GitHub (https://github.com/dekkerlab/cworld-dekker). Compartment changes in Fig EV3 were calculated by taking reproducible compartments across the biological replicates in each experimental sample, and comparing them to the reproducible compartments with their respective controls. The saddle plots were generated by ranking the first eigenvalues and binning them 30 quantiles. Then, the distance-normalized observed/expected interaction frequency between each bin was calculated in a pairwise fashion.

## Data availability

The sequencing data have been deposited in the Gene Expression Omnibus (GEO) under the accession number GSE114337 (https://www.ncbi.nlm.nih.gov/geo/query/acc.cgi?acc=GSE114337).

**Expanded View** for this article is available online.

## Acknowledgements

We would like to thank members of the Rinn and Blencowe Labs for helpful discussion, Chiara Gerhardinger and Thomas Gonatopoulos-Pournatzis for critical reading of the manuscript, Philipp G. Maass for access to the microscopy resources, and Ulrich Braunschweig for assistance with data analysis and feedback. This study was funded by NIH grants U01 DA040612-01 and PO1 GM099117 (JLR), and by a CIHR Foundation Grant (BJB). JLR is the Leslie Orgel Professor of Biochemistry and Howard Hughes Medical Institute Faculty Scholar. BJB holds the University of Toronto Banbury Chair in Medical Research. ARB is a Banting Postdoctoral Fellow through Natural Sciences and Engineering Research Council of Canada (NSERC).

## Author contributions

ARB and JLR devised the project, with input from BJB. ARB performed all the experiments and analyses. ARB, BJB, and JLR wrote the manuscript.

## Conflict of interest

The authors declare that they have no conflict of interest.

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
