## [Review Process File · EMBO Reports]

Differential Contribution of Steady-State RNA and Active Transcription in Chromatin Organization

A. Rasim Barutcu, Benjamin J. Blencowe, John L. Rinn

Review timeline:	Submission date:	12 March 2019
	Editorial Decision:	7 May 2019
	Resubmission:	18 June 2019
	Editorial Decision:	2 August 2019
	Revision received:	5 August 2019
	Accepted:	6 August 2019

Editor: Esther Schnapp

Transaction Report:

1st Editorial Decision

7 May 2019

Thank you for your patience while your manuscript was peer-reviewed at EMBO reports. We have finally received the full set of referee comments that is pasted below.

As you will see, all referees acknowledge that the findings are interesting and should be published. They also have a few suggestions for how the study should be further strengthened and improved.

Given these constructive comments, we would like to invite you to revise your manuscript with the understanding that the referee concerns must be fully addressed and their suggestions taken on board. Please address all referee concerns in a complete point-by-point response. Acceptance of the manuscript will depend on a positive outcome of a second round of review. It is EMBO reports policy to allow a single round of revision only and acceptance or rejection of the manuscript will therefore depend on the completeness of your responses included in the next, final version of the manuscript.

Revised manuscripts should be submitted within three months of a request for revision; they will otherwise be treated as new submissions. Please contact me if a 3-months time frame is not sufficient for the revisions so that we can discuss this further. You can either publish the study as a short report or as a full article. For short reports, the revised manuscript should not exceed 27,000 characters (including spaces but excluding materials & methods and references) and 5 main plus 5 expanded view figures. The results and discussion sections must further be combined, which will help to shorten the manuscript text by eliminating some redundancy that is inevitable when discussing the same experiments twice. For a normal article there are no length limitations, but it should have more than 5 main figures and the results and discussion sections must be separate. In both cases, the entire materials and methods must be included in the main manuscript file.

Supplementary figures, tables and movies can be provided as Expanded View (EV) files, and we can offer a maximum of 5 EV figures per manuscript. EV figures are embedded in the main manuscript text and expand when clicked in the html version. Additional supplementary figures will need to be

included in an Appendix file. Tables can either be provided as regular tables, as EV tables or as Datasets. Please see our guide to authors for more information.

Regarding data quantification, please specify the number "n" for how many independent experiments were performed, the bars and error bars (e.g. SEM, SD) and the test used to calculate p-values in the respective figure legends. This information must be provided in the figure legends. Please also include scale bars in all microscopy images.

Primary datasets produced in this study need to be deposited in an appropriate public database (see <http://msb.embopress.org/authorguide#dataavailability>). Please remember to provide a reviewer password if the datasets are not yet public. The accession numbers and database should be listed in a formal "Data Availability" section (placed after Materials & Method) (see also <http://emboj.embopress.org/authorguide#dataavailability>).

We now strongly encourage the publication of original source data with the aim of making primary data more accessible and transparent to the reader. The source data will be published in a separate source data file online along with the accepted manuscript and will be linked to the relevant figure. If you would like to use this opportunity, please submit the source data (for example scans of entire gels or blots, data points of graphs in an excel sheet, additional images, etc.) of your key experiments together with the revised manuscript. Please include size markers for scans of entire gels, label the scans with figure and panel number, and send one PDF file per figure.

- a complete author checklist, which you can download from our author guidelines (<http://embor.embopress.org/authorguide#revision>). Please insert page numbers in the checklist to indicate where in the manuscript the requested information can be found. The completed author checklist will also be part of the RPF (see below).
- a letter detailing your responses to the referee comments in Word format (.doc)
- a Microsoft Word file (.doc) of the revised manuscript text
- editable TIFF or EPS-formatted figure files in high resolution. In order to avoid delays later in the process, please read our figure guidelines before preparing your manuscript figures at: http://www.embopress.org/sites/default/files/EMBOPress_Figure_Guidelines_061115.pdf

I look forward to seeing a revised version of your manuscript when it is ready. Please let me know if you have questions or comments regarding the revision.

REFeree REPORTS

Referee #1:

This manuscript investigated a potential role of RNA in initiating and/or maintaining high order chromatin interactions, as measured by Hi-C. The basic finding is that once the chromatin architecture is established, RNA does not seem to be needed for its maintenance based on the effect of RNase A treatment before fixing the cell. The authors also examined the Hi-C structure after blocking the overall transcription activity with Act. D, which resulted in a degree of obscured TAD

boundaries, indicating increased interactions between TADs. This suggests a degree of chromatin reorganization, but it remains unclear whether this is due to the disrupted process of transcription or the production of nascent RNA.

Overall, the information presented in this work would be useful to the genomics community, emphasizing that the majority of genomic interactions detected by Hi-C is stationary, rather than dynamic and thus regulatory. I have a few suggestions for the authors to improve the manuscript.

1. After permeabilization, the nuclei became dramatically shrunk before fixation, and with the addition of RNase A, the nuclei were significantly enlarged. This needs to be discussed. There is also a big hole in DAPI-staining nucleus. Is it an expanded nucleolus?
2. Interestingly, such dramatic change in nuclear morphology is not related to altered TAD configuration. However, their data indicate significant disruption of B compartments, as shown in Fig. 2A and S2B. This may be responsible for general decompaction of heterochromatin, which may be related to expanded nucleus. The authors need to describe this effect, especially with respect to the selective effect of RNase A treatment of B compartments.
3. It is a bit strange to measure apoptosis after RNase A treatment. It is unlikely a natural cellular response when cells were permeabilized or treated with RNase A. Moreover, Annexin V is part of cell membrane, and as a result of Tween-20 treatment, it will get lost. Thus, negative staining of Annexin V is not indicative of the absence of apoptosis. More strangely, after RNase A treatment before fixation, more Annexin signals were detected. How did such treatment cause permeabilized cells to induce apoptosis?
4. Act. D treatment clearly caused blurred TAD boundaries. This likely resulted from the lost insulators in combination with disrupted promoter-enhancer interactions. There are two points that can be further made here. First, this may result from disrupted transcription process or diminished RNA, which needs to be discussed. More importantly, the authors may use the existing ChIP-seq data for CTCF and chromatin markers to determine whether specific promoter-enhancer interactions are largely preserved or non-specific interactions around TAD boundaries are increased at the expense of specific promoter-enhancer interactions in the absence of transcription. These additional analyses would enrich the scientific content of the manuscript.

Referee #2:

Many recent studies have instigated a role for RNA / transcription in regulating chromatin structure. However, to dissect the role of RNA per se and transcription per se is extremely difficult, if not impossible. In this study Barutcu et al have attempted to ask how RNA and transcription are important for TAD formation and the organisation of A and B compartments. This is an extremely difficult question and the approaches to investigate it are consequently crude. Concomitantly Barutcu et al have had to use extensive RNase A treatment or actinomycin D treatment to degrade RNAs or inhibit transcription. Both of the approaches are inherently crude, however taking the data at face value they will provide insight into the role of RNA / transcription in regulating chromatin structure (as measured by HiC).

Sensibly the authors have not over interpreted their data but make the useful observation that RNA per se does not have a major impact on the formation of TADs but transcription does affect TADs. In contrast they find that RNase treatment promotes a subtle change in A and B compartment formation. It is difficult to interpret what these changes mean for cellular function but if nothing else they support a role for both RNA and transcription in regulating different level of genome organisation.

The experiments look carefully undertaken and sensibly interpreted and will make a useful addition to the literature.

Referee #3:

Manuscript by Barutcu et al addresses important and poorly understood questions about the roles of RNA molecules and the process of transcription in maintaining 3D genome organization. This is a very important study, as it reports largely "negative" results; our field needs more such findings published. While the experimental approach is sound and has important controls, data analysis is incomplete to support the main claims. Much more thorough analysis needs to be done to reevaluate and support or reject authors claims. My general impression is that chromosome organization remained largely unchanged in presented experiments. Even if this is the case, these are important findings that need to be published.

1. Authors report, but don't highlight, an observation that compartmentalization gets weaker upon RNase A treatment. Judging from Fig 3, it looks like a similar weakening of compartments may be present upon ActD treatment. In both cases, authors need to compute and present "saddle plots" (like in many papers from the Dekker/Mirny) which quantify the degree of compartmentalization by computing the contact frequency as a function of the rank of the eigenvector. Computing eigenvectors and saddle-plots from trans-chromosomal data can help to delineate changes in TADs and in compartments.
2. Weakening of compartments, however, may be caused by large changes in nuclear volumes (shrinkage in bXL Control and swelling in bXL RNase) as seen in Fig S1. It would be great if authors can address this issue experimentally; I'm not sure about experimental strategies to control this. At least mentioning this is important.
3. Figure 2 beautifully shows Hi-C data at multiple scales. Fig 3, unfortunately focuses only on the near-diagonal region; panels similar to Fig 2A need to be presented, alongside with the "saddle plots" and the contact probability $P(s)$ curves.
4. Plots of the contact probability with distance, $P(s)$ curves, can also help to quantify changes between conditions. Moreover, they can be informative to detect changes in the activity of the loop extrusion process, which is reflected in the shoulder at small ($s < 300\text{Kb}$) distances (e.g. PMID: 29728444)
5. Data presented on Fig 3 didn't convince me that TADs get weaker upon ActD treatment. (1) mildly weaker insulation can be due to changes in compartments (e.g. weakening of transcriptionally active A compartment upon transcription inhibition). Insulation score captures both types of boundaries. Changes in compartments in cis and trans need to be quantified (see above) to rule this out (or to support this). (2) To better quantify changes at TAD organization authors can use CTCF ChIPseq data for cell types they used and look at insulations specifically at CTCF peaks. (3) Another way to quantify changes in TAD organization is to examine corner-peaks of TADs (aka "loops" or "dots"). They have been called for K562 in Rao 2014. Authors can examine changes at these corner peaks upon treatments. Changes authors observe in Fig 3 may be in part due to alterations $P(s)$ curve, examine these curves as well as plotting insulation as obs/exp (with expected from $P(s)$) can shed light on this issue.
6. In one place the paper claims to examine correlations between Hi-C maps. Pearson correlations between Hi-C maps are dubious for assessing reproducibility, since it mostly captures the steep scaling of contact frequency with genomic distance. One can at least use obs/exp, or better, correlate PCA eigenvectors and insulation score tracks between replicate experiments.
7. The paper mentioned "single stranded RNA". Authors need to clarify whether they targeted single stranded RNA or all RNAs in their RNase treatments.
8. Discussion: "Based on these results, we hypothesize that TAD boundaries form during early mitosis". It's unclear what they are referring to as TADs disappear in prophase. Moreover, Vian et al have demonstrated that TADs can make disappear (upon cohesin degradation) and reappear (upon auxin washoff) even in the absence of transcription. These results clearly show that transcription is not essential for establishing TADs.

Leonid Mirny

Referee 1:

This manuscript investigated a potential role of RNA in initiating and/or maintaining high order chromatin interactions, as measured by Hi-C. The basic finding is that once the chromatin architecture is established, RNA does not seem to be needed for its maintenance based on the effect of RNase A treatment before fixing the cell. The authors also examined the Hi-C structure after blocking the overall transcription activity with Act. D, which resulted in a degree of obscured TAD boundaries, indicating increased interactions between TADs. This suggests a degree of chromatin reorganization, but it remains unclear whether this is due to the disrupted process of transcription or the production of nascent RNA.

Overall, the information presented in this work would be useful to the genomics community, emphasizing that the majority of genomic interactions detected by Hi-C is stationary, rather than dynamic and thus regulatory. I have a few suggestions for the authors to improve the manuscript.

We thank the reviewer for their positive feedback and constructive comments, for which we provide a point-by-point response below.

1. After permeabilization, the nuclei became dramatically shrunk before fixation, and with the addition of RNase A, the nuclei were significantly enlarged. This needs to be discussed. There is also a big hole in DAPI-staining nucleus. Is it an expanded nucleolus?

We agree with the reviewer that the observed changes in nuclear volume prior to fixation and following RNase A treatment are important to mention and discuss. To address this, we have performed additional microscopy experiments and analyses to better characterize differences in nuclear size.

First, to address the reviewer's comments about nucleolar morphology, we performed immunostaining for the nucleolar marker Fibrillarin on the RNase- or ActD-treated cells, as well as on control cells. Since K562 cells are small, we used the Zeiss LSM880-Airyscan system, which enhances sensitivity and resolution beyond the diffraction limit of light. We imaged whole cells using z-stacks (with 0.45 micron slices) and presented the Maximum Intensity Projection of the images. We observe similar Fibrillarin immunostaining patterns between the bXL RNase A and control treatments. A figure showing these data is included in Supplementary Figure S1B. These analyses are described in the Results and interpreted in the Discussion.

Second, regarding the reviewer's point concerning the apparent expanded nuclei after RNase treatment, we quantified nuclear sizes of bXL CTRL and bXL RNase A K562 cells and observed similar ranges of nuclear sizes across the conditions (Supplementary Figure S1D). It worth noting that this result is concordant with a previous study (Caudron-Herger et al., Nucleus, 2011). That RNase treatment has relatively little impact on overall nuclear morphology, assessed by calculating nuclear sizes and by Fibrillarin staining, regardless of whether or not prior cross-linking was performed, is now mentioned in the revised text (Results page 4 and Discussion, page 9).

2. Interestingly, such dramatic change in nuclear morphology is not related to altered TAD configuration. However, their data indicate significant disruption of B compartments, as shown in Fig. 2A and S2B. This may be responsible for general decompaction of heterochromatin, which may be related to expanded nucleus. The authors need to describe this effect, especially with respect to the selective effect of RNase A treatment of B compartments.?

As requested, we have elaborated on this observation in the Discussion (page 9). We also refer the reviewer to our response to Reviewer #3 below, for which we performed several additional bioinformatic analyses revealing that the bXL RNase A samples harbor perturbed B-type compartmental interactions.

As mentioned in response to the previous point by this reviewer, we also performed additional morphological and immunofluorescence microscopy characterization of several properties of nuclear morphology (Supplementary Figure S1), thus, providing a more detailed understanding of

the treatments and conditions analyzed by Hi-C as a relative benchmark, as suggested by the reviewer.

Also as suggested by the reviewer, we have placed these results in the context of A-B compartmentalization changes. Specifically, we have moved the new data mentioned by the reviewer into Figure 2C-B and Figure 4B. Moreover, we have simplified the text to forge a better connection to remaining questions about nuclear morphology. Overall, despite the crude nature of treatments, they result in subtle effects to A-B compartmental organization and, as mentioned above, this is the main conclusion we wish to convey.

3. It is a bit strange to measure apoptosis after RNase A treatment. It is unlikely a natural cellular response when cells were permeabilized or treated with RNase A. Moreover, Annexin V is part of cell membrane, and as a result of Tween-20 treatment, it will get lost. Thus, negative staining of Annexin V is not indicative of the absence of apoptosis. More strangely, after RNase A treatment before fixation, more Annexin signals were detected. How did such treatment cause permeabilized cells to induce apoptosis?

We understand the reviewer's comments. However, based on initial feedback from colleagues, the question was raised as to whether there may be a limited apoptotic response following the treatments that could result in altered chromatin architecture, which is why we performed Annexin V immunostaining. Given the complications with using Annexin V immunostaining raised by the reviewer, we have replaced these data with a new microscopy analysis showing immunostaining for activated Caspase 3, an apoptotic marker that is localized in the nucleus and cytoplasm. Consistent with the results of Annexin V staining, we observe weak but comparable levels of active Caspase 3 immunostaining. This result is consistent with a previous report (Caudron-Herger et al., Nucleus, 2011), that RNase A microinjected cells display an apoptotic response at later times (~1 hour after) the nuclear changes occur. These data are included in Supplementary Figure S1A.

4. Act. D treatment clearly caused blurred TAD boundaries. This likely resulted from the lost insulators in combination with disrupted promoter-enhancer interactions. There are two points that can be further made here. First, this may result from disrupted transcription process or diminished RNA, which needs to be discussed. More importantly, the authors may use the existing ChIP-seq data for CTCF and chromatin markers to determine whether specific promoter-enhancer interactions are largely preserved or non-specific interactions around TAD boundaries are increased at the expense of specific promoter-enhancer interactions in the absence of transcription. These additional analyses would enrich the scientific content of the manuscript.

Based on the reviewer's insightful suggestion, we sought to ask whether CTCF binding is associated with the reduction in TAD boundary scores upon 24hr of ActD treatment. To address this question, we intersected the ENCODE K562 CTCF ChIP-seq data with Hi-C data by categorizing the 40kb TAD boundary bins as either "bound by 1 or less CTCF sites", or "bound by 2 or more sites". Consistent with earlier findings, we observed significantly higher scores of TAD boundaries that harbored 2 or more CTCF sites in both control and 24 hour ActD treated cells (new Figure 5E, shown below), which are stronger and have higher insulation scores. Interestingly, however, ActD treatment led to a significant decrease in TAD boundary scores compared to controls regardless of the number of CTCF sites bound. These findings suggest that the act of transcription may affect TAD boundaries independent of CTCF. These findings are mentioned in the 3rd paragraph of our Discussion section on page 8.

Figure 5E: Boxplot showing the TAD boundary scores that are either bound by ≤ 1 or ≥ 2 CTCF sites in control (0hr) and 24hr ActD treated (24hr) cells. Even though the TAD boundaries with more number of CTCF sites have significantly higher TAD boundary scores (p : Wilcoxon rank-sum test), the 24hr sample displayed lower TAD boundary scores, regardless of the number of CTCF sites bound.

Also, based on Reviewer #3's suggestion, we performed an Aggregate Peak Analysis based on the K562 "loops" detected in a high-resolution Hi-C study (Rao et al., Cell, 2014), and observed that the loop contacts show a $\sim 50\%$ decrease (by z-score and log₂ fold change) in ActD-treated cells compared to controls.

We believe that these two additional analyses strongly support the conclusion that transcriptional inhibition leads a reduction in TAD boundary strength.

Figure 5D: Aggregate Peak Analysis of control and 24hr ActD treated cells. The ActD treated cells show $\sim 50\%$ decrease of loop intensity by z-score and the log₂ fold change.

The findings of this analysis have now been added to the Results, and their interpretation is incorporated in the revised Discussion.

We thank the reviewer for their valuable comments, which have helped us to significantly improve the quality of our manuscript.

Referee 2:

Many recent studies have instigated a role for RNA / transcription in regulating chromatin structure. However, to dissect the role of RNA per se and transcription per se is extremely difficult, if not

impossible. In this study Barutcu et al have attempted to ask how RNA and transcription are important for TAD formation and the organisation of A and B compartments. This is an extremely difficult question and the approaches to investigate it are consequently crude. Concomitantly Barutcu et al have had to use extensive RNase A treatment or actinomycin D treatment to degrade RNAs or inhibit transcription. Both of the approaches are inherently crude, however taking the data at face value they will provide insight into the role of RNA / transcription in regulating chromatin structure (as measured by HiC).

Sensibly the authors have not over interpreted their data but make the useful observation that RNA per se does not have a major impact on the formation of TADs but transcription does affect TADs. In contrast they find that RNase treatment promotes a subtle change in A and B compartment formation. It is difficult to interpret what these changes mean for cellular function but if nothing else they support a role for both RNA and transcription in regulating different level of genome organisation.

The experiments look carefully undertaken and sensibly interpreted and will make a useful addition to the literature.

We thank the reviewer for his/her positive comments and support of our manuscript.

Referee #3:

Manuscript by Barutcu et al addresses important and poorly understood questions about the roles of RNA molecules and the process of transcription in maintaining 3D genome organization. This is a very important study, as it reports largely "negative" results; our field needs more such findings published. While the experimental approach is sound and has important controls, data analysis is incomplete to support the main claims. Much more thorough analysis needs to be done to reevaluate and support or reject authors claims. My general impression is that chromosome organization remained largely unchanged in presented experiments. Even if this is the case, these are important findings that need to be published.

We thank the reviewer for the thoughtful review of our manuscript and helpful suggestions for revisions, which are addressed below.

1. Authors report, but don't highlight, an observation that compartmentalization gets weaker upon RNase A treatment. Judging from Fig 3, it looks like a similar weakening of compartments may be present upon ActD treatment. In both cases, authors need to compute and present "saddle plots" (like in many papers from the Dekker/Mirny) which quantify the degree of compartmentalization by computing the contact frequency as a function of the rank of the eigenvector. Computing eigenvectors and saddle-plots from trans-chromosomal data can help to delineate changes in TADs and in compartments.

To address the reviewer's concern, in Figure 2E we present a violin plot showing the 1st eigenvalues (EV), which are calculated using the trans-interactions of the control and RNase treated samples before and after crosslinking (see below). It can be inferred from the bXL RNase A plot that that there is a reduction of the negative eigenvalues, suggesting a perturbation of the B-type compartments.

Figure 2E: Violin plots showing the 1st eigenvalue, calculated by using the trans-chromosomal data of RNase treated and control cells before and after crosslinking. The bXL RNase A samples harbors a more prominent reduced bin density with negative eigenvalues, suggesting a perturbation of B-type compartmentalization. *p*-value: Wilcoxon rank-sum test.

A similar plot was generated for transcriptionally inhibited cells, (Figure 4C, below), which suggests a preservation of compartmentalization upon transcription inhibition.

Figure 4C: Violin plots showing the 1st eigenvalue calculated using the trans-chromosomal data of ActD treated and control cells. *p*-value: Wilcoxon rank-sum test.

In addition, as per the reviewer's request, we generated the suggested saddle plots by ranking the 1st eigenvalues, and then by binning them into 30 quantiles, followed by calculating the pairwise distance-normalized observed/expected interaction frequencies among the 30 quantile bins. We generated these plots both for the cis-contacts (*intra*-chromosomal), as well as for the trans-contacts (*inter*-chromosomal). The saddle plots for RNase treated cells are presented in Figure 2F (see below).

As pointed out by the reviewer, we indeed observe a reduction of specifically B-B interactions in saddle plots generated with both in-cis and in-trans chromosomal data (Figure 2F). These results

suggest that RNase A treatment before crosslinking, although not affecting TAD structures (Figure 3), leads to global weakening of compartmentalization and perturbations of B-type compartments.

The same analysis for the transcriptionally inhibited cells (Figure 4D), however, did not reveal any notable differences aside from subtle changes in A-B and B-A interactions (please see comments below). However, we also note that these effect-sizes are small relative to the gross-level of treatments, and as such we have taken care not to overstate this aspect of the study.

Figure 2F: Saddle plots showing the compartmental interactions in RNase treated and control cells before and after crosslinking. Bins were assigned to 30 quantiles based on their PC1 scores, and the average distance-normalized observed/expected interaction scores for each pair were calculated. The bXL RNase sample show reduced B-B interactions.

2. Weakening of compartments, however, may be caused by large changes in nuclear volumes (shrinkage in bXL Control and swelling in bXL RNase) as seen in Fig S1. It would be great if authors can address this issue experimentally; I'm not sure about experimental strategies to control this. At least mentioning this is important.

We agree that this is an important point that is challenging to address. Based also on Reviewer #1's comments, we have performed additional microscopy to better characterize changes in nuclear morphology and organization in K562 cells treated with Tween-20 and RNase A before crosslinking. Please refer to Reviewer #1 for details (shown below in *italics*).

“We agree with Reviewer #1 that the observed changes in nuclear volume prior to fixation and following RNase A treatment are important to mention and discuss. To address this, we have performed additional microscopy experiments and analyses to better characterize differences in nuclear size.

First, to address the reviewer's comments about nucleolar morphology, we performed immunostaining for the nucleolar marker Fibrillarin on the RNase- or ActD-treated cells, as well as on control cells. Since K562 cells are small, we used the Zeiss LSM880-Airyscan system, which enhances sensitivity and resolution beyond the diffraction limit of light. We imaged whole cells using z-stacks (with 0.45 micron slices) and presented the Maximum Intensity Projection of the images. We observe similar Fibrillarin immunostaining patterns between the bXL RNase A and control treatments. A figure showing these data is included in Supplemental Figure S1B. These analyses are described in the Results and interpreted in the Discussion.

Second, regarding the reviewer's point concerning the apparent expanded nuclei after RNase treatment, we quantified nuclear sizes of bXL CTRL and bXL RNase A K562 cells and observed similar ranges of nuclear sizes across the conditions (Supplementary Figure S1D). It worth noting

that this result is concordant with a previous study (Caudron-Herger et al., *Nucleus*, 2011). That RNase treatment has relatively little impact on overall nuclear morphology, assessed by calculating nuclear sizes and by Fibrillarin staining, regardless of whether or not prior cross-linking was performed, is now mentioned in the revised text (Results page 4 and Discussion, page 9)."

3. Figure 2 beautifully shows Hi-C data at multiple scales. Fig 3, unfortunately focuses only on the near-diagonal region; panels similar to Fig 2A need to be presented, alongside with the "saddle plots" and the contact probability P(s) curves.

We have included the Hi-C matrices at multiple resolutions, showing regions off the diagonal for the ActD treated and control cells in Figure 4A (please see the below figure). Even at 100kb resolution, the ActD treated samples appear to have a "fuzzier" diagonal, indicative of weakened TAD boundary formation.

A

Figure 4A: Hi-C interaction matrices at multiple resolutions for control (0hr) and 24hr ActD treated cells.

In addition, we generated compartmental saddle plots by using the same approach explained earlier (Figure 4D, below), and the P(s) curve, for large-distance interactions (Figure 4E, shown below), for the ActD treated and control cells. We observed similar B-B and A-A compartmentalization,

however there appears to be a subtle difference in A-B and B-A interactions, which may be reflected as reduced Pearson correlations in Figure 4B.

Figure 4D: Saddle plots showing the compartmental interactions for both cis- and trans- contacts in control and 24hr ActD treated cells. The samples display similar patterns of compartmental interactions.

Figure 4E: Scaling plot generated by using the 500kb resolution Hi-C data for control and 24hr ActD treated cells. The samples show similar decay rates.

4. Plots of the contact probability with distance, $P(s)$ curves, can also help to quantify changes between conditions. Moreover, they can be informative to detect changes in the activity of the loop extrusion process, which is reflected in the shoulder at small ($s < 300\text{Kb}$) distances (e.g. PMID: 29728444)

We agree with the reviewer and thus have included the $P(s)$ curve generated with 40kb Hi-C data with a distance limit of 3Mb. We also have included the suggested citation in our references. In the figure below, indeed, the 24hr ActD treated sample shows higher interaction frequencies at shorter distances (i.e the rate of decay is lower in distances at $\sim 100\text{kb}$ range). This figure is presented in Figure 5F. We thank the reviewer for the valuable suggestion to include this plot.

Figure 5F: Scaling plot showing the interaction frequency as a function of genomic distance in control and 24hr ActD treated cells.

5. Data presented on Fig 3 didn't convince me that TADs get weaker upon ActD treatment. (1) mildly weaker insulation can be due to changes in compartments (e.g. weakening of transcriptionally active A compartment upon transcription inhibition). Insulation score captures both types of boundaries. Changes in compartments in cis and trans need to be quantified (see above) to rule this out (or to support this). (2) To better quantify changes at TAD organization authors can use CTCF ChIPseq data for cell types they used and look at insulations specifically at CTCF peaks. (3) Another way to quantify changes in TAD organization is to examine corner-peaks of TADs (aka "loops" or "dots"). They have been called for K562 in Rao 2014. Authors can examine changes at these corner peaks upon treatments. Changes authors observe in Fig 3 may be in part due to alterations P(s) curve, examine these curves as well as plotting insulation as obs/exp (with expected from P(s)) can shed light on this issue.

The reviewer raises important suggestions and we address each one of his points below:

As stated above, we generated saddle plots by using the cis- as well as the trans- data, however did not find a notable difference, apart from the fact that there appears to be subtly more A-B and B-A contacts in ActD treated cells. In addition, in the original submission, we presented the compartmental switching (i.e B to A or A to B) in Supplementary Figure S3, and have noted that >90% of compartmentalization is similar between the control and ActD treatment conditions. This suggests that transcriptional inhibition of all three polymerases does not lead to a significant perturbation of genomic compartmentalization.

Figure 4D: Saddle plots showing compartmental interactions for both cis- and trans- contacts in control and 24hr ActD treated cells. The samples display similar patterns of compartmental interactions.

Based on this reviewer's and Reviewer #1's comments (response to point 4), we sought to ask whether CTCF binding affected the reduction of TAD boundary scores upon 24hr of ActD treatment. To address this question, we intersected the ENCODE K562 CTCF ChIP-seq data with Hi-C by categorizing the 40kb TAD boundary bins as either "bound by 1 or less CTCF sites", or "bound by 2 or more sites". Consistent with earlier findings, we observed significantly higher scores of TAD boundaries that harbored 2 or more CTCF sites in both control and 24hour ActD treated cells (new Figure 6E, shown below), which are stronger and have higher insulation scores. Interestingly, however, ActD treatment led to a significant decrease of TAD scores compared to controls regardless of the number of CTCF sites bound, suggesting that the act of transcription may have roles / effects on TAD boundaries independent of CTCF.

Figure 5E: Boxplot showing the TAD boundary scores that are either bound by ≤ 1 or ≥ 2 CTCF sites in control (0hr) and 24hr actinomycin D treated (24hr) cells. Even though the TAD boundaries with more number of CTCF sites have significantly higher TAD boundary scores (p : Wilcoxon rank-sum test), the 24hr sample displayed lower TAD boundary scores, regardless of the number of CTCF sites bound.

Based on the reviewer's suggestion, we performed an Aggregate Peak Analysis on control and 24hr ActD treated cells, by using the Rao et al. 2014 K562 loop coordinates. Consistent with the reduction in insulation scores, we identified a $\sim 50\%$ decrease in the loop strength (assessed by the z-score and the \log_2 change). The Figure can be found below and is now presented as Figure 5D.

Figure 6D: Aggregate Peak Analysis for control (0hr) and 24hr ActD treated cells. There is a ~50 reduction in loop intensity upon transcriptional inhibition when compared to controls, correlating with the weakening of TAD borders.

We believe that these analyses further support our data suggesting global weakening of TAD boundaries independent of compartmentalization, upon transcriptional inhibition.

6. In one place the paper claims to examine correlations between Hi-C maps. Pearson correlations between Hi-C maps are dubious for assessing reproducibility, since it mostly captures the steep scaling of contact frequency with genomic distance. One can at least use obs/exp, or better, correlate PCA eigenvectors and insulation score tracks between replicate experiments.

The reviewer is correct in his point about the replicate correlations. However, the replicate correlations presented in Supplementary Figure S2 were already generated by using the PCA 1st eigenvectors, as the reviewer suggested. We apologize for not making this clear before and have now added the missing information to the Results and Methods section of the manuscript.

7. The paper mentioned "single stranded RNA". Authors need to clarify whether they targeted single stranded RNA or all RNAs in their RNase treatments.

RNase A is an endoribonuclease which specifically cleaves single-stranded RNA. We have clarified this in the Introduction.

8. Discussion: "Based on these results, we hypothesize that TAD boundaries form during early mitosis". It's unclear what they are referring to as TADs disappear in prophase. Moreover, Vian et al have demonstrated that TADs can make disappear (upon cohesin degradation) and reappear (upon auxin washoff) even in the absence of transcription. These results clearly show that transcription is not essential for establishing TADs.

The point in the Discussion we intended to make is that the loop extrusion complex is the sole factor responsible for establishing the TADs, as also supported literature evidence.

However, the strength of TAD boundaries, which may be exploited by the cells as a means of chromatin/transcriptional regulation, can be modified (i.e fine-tuned) by the act of transcription, and or binding of multiple chromatin binding factors. Thus, our intention is not to argue the essentiality of transcription for TAD formation, but merely suggest it could be involved in the fine-tuning of TAD boundary strength. We have revised the Discussion to better reflect this interpretation, specifically emphasizing that TADs can disappear and reappear in the absence of transcription, and that transcription is not essential for establishing TADs.

Overall, we agree with the reviewer's comments and his suggestions have helped us to significantly strengthen the conclusions of our manuscript. However, it is important to note that the most surprising result of this study is the lack of change in TAD boundary strength upon RNase treatment. Combined with the more ambiguous definition of A-B compartmentalization, we however have taken considerable care in how we have interpreted the results given the crude nature of treatments used in our experiments.

Leonid Mirny

We thank Dr. Mirny for his extensive and constructive comments on our manuscript.

2nd Editorial Decision

2 August 2019

Thank you for the submission of your revised manuscript, and I am truly sorry for this unusual delay in its handling process. We still do not understand why this manuscript disappeared from our system. I have now received the comments from referee 1 who supports the publication of your work, and since referee 3 is not responsive, I decided that we will publish your manuscript pending only minor formal revisions described below.

Please upload all figures as individual files.

The data availability section with the accession numbers needs to be moved to the end of the methods section.

I attach a word file with comments to this email. Please address all comments and upload the corrected manuscript file with your final submission.

The supplementary figures need to be called Expanded View (EV) figures. EV Figures should be cited as 'Figure EV1, Figure EV2' etc... in the text and their respective legends should be included in the main text after the legends of the main figures. Please also provide all info on statistics in the EV legends, as for the main figures.

The table with the sequencing statistics should be called Table EV1, and the table with the primer sequences could be a regular table in the methods section.

Please remember to cite all main and all EV figures and EV tables somewhere in the manuscript text. Currently, Fig 4E and 5B are not called out, please correct.

For our website, we need a synopsis image (you already sent an image, and we can use that), a short summary of the findings and their significance, and 2-3 bullet points highlighting key results.

Please reduce the number of keywords to 5.

I look forward to seeing a final version of your manuscript as soon as possible. Let me know if you have any comments or questions.

REFeree REPORT

Referee #1:

The authors have addressed all of my previous questions as well as those from the other two reviewers, and revised the manuscript accordingly. I am fine with accepting the manuscript.

The authors performed all minor editorial changes.

Corresponding Author Name: A. Rasim Barutcu & John L. Rinn

Manuscript Number: EMBOR-2019-48068V1